# Effect of Bioactive Glass-Based Root Canal Sealer on the Incidence of Postoperative Pain after Root Canal Obturation

**DOI:** 10.3390/ijerph17238857

**Published:** 2020-11-28

**Authors:** Ayako Washio, Hiroki Miura, Takahiko Morotomi, Miki Ichimaru-Suematsu, Hirotake Miyahara, Kaori Hanada-Miyahara, Shinji Yoshii, Kazumasa Murata, Nana Takakura, Eiichi Akao, Masataka Fujimoto, Atsushi Matsuyama, Chiaki Kitamura

**Affiliations:** 1Division of Endodontics and Restorative Dentistry, Department of Oral Functions, Kyushu Dental University, 2-6-1 Manazuru, Kokurakita-ku, Kitakyushu, Fukuoka 803-8580, Japan; r13morotomi@fa.kyu-dent.ac.jp (T.M.); r11ichimaru@fa.kyu-dent.ac.jp (M.I.-S.); r08yoshii@fa.kyu-dent.ac.jp (S.Y.); r17murata@fa.kyu-dent.ac.jp (K.M.); r18akao@fa.kyu-dent.ac.jp (E.A.); r12fujimoto@fa.kyu-dent.ac.jp (M.F.); r06kitamura@fa.kyu-dent.ac.jp (C.K.); 2Miura Dental Clinic, 215-2 Oazakusagi, Omuta, Fukuoka 837-0917, Japan; r17miura@fa.kyu-dent.ac.jp; 3Shimata Dental Clinic, 1-2-17 Shimata, Hikari, Yamaguchi 743-0063, Japan; miyacchi825@gmail.com; 4Hanada Dental Clinic, 1-8-21 Sakae, Kudamatsu, Yamaguchi 744-0013, Japan; hanakao0531@gmail.com; 5Medical Corporation Fukuwakai Beppu Dental Clinic, 4-27-1 Chihaya, Higashi-ku, Fukuoka 813-0044, Japan; r15takakura@outlook.jp (N.T.); a.matsuyama@k-showakai.net (A.M.)

**Keywords:** bioactive glass-based sealer, bioceramics, biocompatibility, postoperative pain, root canal obturation

## Abstract

The purpose of this study is to evaluate the effect of a bioactive glass-based root canal sealer, Nishika Canal Sealer BG (CS-BG), on the incidence of postoperative pain (PP) after root canal obturation (RCO). Eleven dentists performed pulpectomy or infected root canal treatments for 555 teeth. During RCO, CS-BG was used. After RCO, the rate of PP and the factors affecting PP (pain during RCO and pain immediately after RCO) were analyzed. PP was observed in eight teeth (1.5%), and within 7 days after RCO, there were no teeth with pain. In these teeth with PP, there was a significant difference in the occurrence of pain during RCO, but not in the occurrence of pain immediately after RCO, when compared with pulpectomy and infected root canal treatment. These clinical results show that CS-BG has an excellent biocompatibility, and can suppress the distress of patients during RCO.

## 1. Introduction

Recently published systematic reviews reported that postoperative pain (PP) in endodontics ranges between 1.5 and 58%, as described in different studies [1,2,3]. The presence of PP is associated with a number of endodontic treatment-related parameters, including the estimation of working length with an apex locator connected to every file [4], number of visits [5], choice of instrumentation [6], and choice of root canal sealer [7]. Particularly, PP associated with root canal sealer ranges between 5 and 20% [8,9,10].

Root canal sealers placed in the root canals interfere with the periodontal tissues through the apical foramina, lateral canals, or leaching and can potentially affect the healing process in the periodontal tissues. Therefore, root canal sealers with high biocompatibility and high sealing properties are desired. When the root canals are obturated by root canal filling materials, PP is often caused. This PP depends on a number of different factors, including the composition of the sealer [11].

The evolution of sealers started from zinc oxide eugenol-based preparations which have a history of use in root canal obturation (RCO) for over a century, and these are highly acclaimed for their antimicrobial property [12]. However, eugenol is found to elute from zinc oxide eugenol-based sealer, which is known to induce a toxic effect [13]. The effect is persistent even after setting of the material. Hence, non-eugenol-based sealer was developed. Subsequently, the era of contemporary sealers, the resin-based sealers (Resin), such as epoxy-resin- or 4-META-based sealers having properties of adhesion, began [12]. In the recent years, bioceramics-based sealer has been developed for better biocompatibility and strong bonding.

The bioactive glass (BG)-based sealer, Nishika Canal Sealer BG (CS-BG; Nippon Shika Yakuhin Co., Ltd., Yamaguchi, Japan), one of the bioceramic-based sealer, has been developed in 2017 and is now being applied in endodontic treatment. CS-BG demonstrates many desirable properties, such as physicochemical stability, biocompatibility, sealing ability, and removability [14]. Biocompatibility is the ability of a material to achieve a stable and advantageous host response during application [15]. The in vitro and in vivo studies have reported that CS-BG has excellent biocompatibility for the periapical tissues [16,17,18]. However, the clinical significance of these characteristics of CS-BG is still unclear.

The aim of this study was to evaluate the effect of CS-BG on the incidence of PP after RCO.

## 2. Materials and Methods 

### 2.1. Patient Selection

Patients treated in the Division of Tooth Therapy (Endodontics and Restorative Dentistry) of Kyushu Dental University Hospital and a private practice clinic setting, between November 2017 and December 2019, were included in this study. Ethical approval for the study was obtained from the ethical committee of Kyushu Dental University (approval no: 17–39; Kyushu Fukuoka, Japan). Five hundred and fifty-five root canal obturated teeth requiring endodontic treatment performed by 11 dentists (4 endodontists and 5 dentists with more than 5 years of experience, and 2 dentists within 5 years of experience) were selected for this study.

### 2.2. Treatment Protocol

The dentists examined and diagnosed the disease of patients using radiographic and clinical findings, and the same dentist diagnosing a case was assigned for the endodontic treatment. For the pulpectomy, the patient was administered local anesthesia. The anesthetic effect was verified by subjective symptoms (tingling and numbness). In case of severe loss of tooth structure, the pre-endodontic build-up of the crown was prepared for facilitating the rubber dam placement and the temporization of the tooth between visits, and the tooth was isolated using a rubber dam.

Access cavity was prepared using sterile diamond points and carbide burs. The canal patency was checked by #10 K-file (MANI, Inc., Tochigi, Japan) and a glide path was established using #15 K-file (MANI, Inc., Tochigi, Japan). The working length was determined using an electronic apex locator (Root ZX II; Morita, Kyoto, Japan). In the cases in which a reliable electronic apex locator reading could not be achieved, a radiograph was taken to confirm the working length. The canals were irrigated with 3% sodium hypochlorite (NaClO; Nippon Shika Yakuhin Co., Ltd, Yamaguchi, Japan) using an endodontic syringe with a closed-ended side-vented needle after each filing. The master apical file size was case-specific and determined based on the initial canal size. In re-treatment cases, previous root canal filling materials were removed using a combination of ultrasonics, a gutta percha solvent (GP-Solvent; Nippon Shika Yakuhin Co.,Ltd, Yamaguchi, Japan), rotary instruments, and K-files.

After root canal preparation, all the canals were irrigated with ultrasonic activation (P-Max plus; Acteon Inc, France) at two sets of 30 s with 3% NaClO and 3% EDTA solution each [19] (Nippon Shika Yakuhin Co., Ltd., Yamaguchi, Japan), and then, finally, the canals were washed with sterilized saline (Otsuka Pharmaceutical Co., Ltd., Tokyo, Japan). After the completion of irrigation, the canals were dried using endo aspirator and paper points, and were filled with calcium hydroxide paste (Calcipex Plane II; Nippon Shika Yakuhin Co.,Ltd, Yamaguchi, Japan). The coronal cavity was sealed with an intermediate restorative material.

After improvement of the patient’s symptoms, all the root canals were obturated with CS-BG and gutta-percha points (GP; Morita, Kyoto, Japan). CS-BG is a two-phase paste: Paste A consists of fatty acids, bismuth subcarbonate, and silica dioxide and Paste B consists of magnesium oxide, calcium silicate glass (a type of BG), and silicate dioxide. CS-BG was prepared according to the manufacturer’s instructions. After irrigation and drying of the canals with root canal vacuum and paper points, a master GP was adapted, and the canals were obturated by single-cone technique (Single), multi-cone technique without condensation (Multi-Non press), lateral condensation technique (Multi-Lateral), or vertical condensation technique (Vertical). The excess sealer and GP were removed using a heated plugger. The coronal cavity was sealed with an intermediate restorative material. The obturation was verified with a radiograph.

### 2.3. Outcome Assessment

Patient attributes with gender, age, arch, type of tooth, symtom, diagnosis, initial- or re-treatment, endodontic treatments (pulpectomy or infected root canal treatment), and root canal obturation techniques Single, Multi-Non press, Multi-Lateral, and Vertical) were examined. Additionally, the incidence of PP after root canal obturation (RCO) was examined. In this study, “PP” was defined as pain induced by root canal sealer during RCO after removing pain on the infection of the tooth by endodontic treatment, and was classified into two kinds of pain: pain during root canal obturation (PD) and pain immediately after root canal obturation (PI). The pain score indicated as “No pain,” “Discomfort,” and “Pain” was recorded to measure PD and PI.

### 2.4. Statistical Analysis

The data were analyzed using the chi-square test, and the significance was set at *p* < 0.05 using the SPSS statistics 25.0 (IBM Corp, Armonk, NY, USA).

## 3. Results

Figure 1 shows the demographic data. A total of 555 teeth were included in this study. One hundred and ninety-five (35%) teeth belonged to men, and 360 (65%) teeth belonged to women (Figure 1a). Ages ranged between 11 and 89 years as follows: 6 (1%), 19 (4%), 54 (10%), 90 (16%), 114 (21%), 117 (21%), 124 (22%), and 31 (5%) teeth were aged in the range of 11–19, 20–29, 30–39, 40–49, 50–59, 60–69, 70–79, and 80–89 years, respectively (Figure 1b). Figure 1c shows the arch and type of tooth. Two hundred and ninety-five (54%) of the treated teeth were in the maxillary, and 260 (46%) teeth were mandibular (Figure 1c). One hundred and sixty-seven (30%) of the treated teeth were located in the anterior segment; 156 (28%) teeth were premolars, and 232 (42%) teeth were molars (Figure 1c). Four hundred and thirty-four (78%) teeth were the positive of symptom, and 121 (22%) teeth were the negative of symptom (Figure 1d). Sixty-four (11%), 62 (11%), 10 (2%), 26 (5%), and 393 (71%) teeth were diagnosed with acute pulpitis, chronic pulpitis, necrosis, acute periodontitis, and chronic periodontitis, respectively (Figure 1e). Two hundred and twelve (38%) teeth underwent initial treatment and 343 (62%) teeth underwent re-treatment (Figure 1f). Pulpectomy was performed on 122 (22%) teeth, and infected root canal treatment was performed on 433 (78%) teeth (Figure 1g). Root canal obturation was done by the single, multi-non press, multi-lateral, and vertical technique for 132 (24%), 307 (55%), 113 (20%), and three (1%) teeth, respectively (Figure 1h).

Figure 2 shows PD and PI. In PD, 525 (95%) teeth perceived “No pain”, and 22 (4%) teeth felt “Discomfort”, and only eight (1%) teeth perceived “Pain” (Figure 2a). In PI, 547 (98.5%) teeth perceived “No pain”, five (1%) teeth felt “Discomfort”, and only three (0.5%) teeth perceived “Pain” (Figure 2b).

Table 1 shows the incidence of PD. There was a significant association between PD and endodontic treatment.

Table 2 shows the incidence of PI. There was no significant difference according to gender, age, arch, type of tooth, type of treatment, endodontic treatment, and root canal obturation technique.

## 4. Discussion

The attribution of the incidence of pain to any specific factor is difficult in clinical research, because endodontic treatment comprises a complex number of procedures, including chemo-mechanical debridement and root canal obturation. Root canal obturation is usually completed in the same appointment, as it has been affirmed that the root canal is the most clean with the least microbial load immediately after chemo-mechanical preparation [20]. The sealer application by the master cone coating and pumping technique is performed to better seal the voids between the obturating material and dentinal walls. Therefore, the sealer and root canal obturation technique may cause the incidence of pain. When designing this study to evaluate pain induced by CS-BG, considerable attention was paid to the control factors that could potentially provoke PD and PI.

In this study, we used the verbal descriptor scale (VDS), but not the visual analog scale (VAS) for postoperative pain assessment. The VAS is a validated method for measuring postoperative pain in dental research [21,22]. However, VAS is originally a validated, subjective measuring method for identifying the changes in the symptoms of acute and chronic pain [23,24]. The evaluation of pain in medical practices (e.g., root canal obturation), which is in small scales, by the VAS may be difficult. Additionally, it is important that the pain assessment scale facilitates the use of instruments with proven reliability and validity in both younger and older patients. A rather consistent finding in the previous studies is that older patients show a higher failure rate on the VAS in comparison with various other pain intensity scales [25,26]. Furthermore, the VDS may be considered when the study population consists of a majority of older patients [27]. Therefore, we evaluated pains with the three scores similar to the VDS, indicated as “No pain,” “Discomfort,” and “Pain”.

Previous studies have indicated that pain after endodontic treatment was particularly related to gender and multirooted teeth [21,28]. Differences between the genders may be explained by differences in physiological reactions to pain [29,30]. However, this study showed that it is likely that sex and the type of tooth did not affect the incidence of PD and PI. Three cases of “Pain” level of PI had perforation or big expansion of root canal apex, indicating that a large contact area between the sealer and periapical tissues physiochemically induced pain. In addition, the influence of a patient’s age was not significantly related to the incidence of PD and PI. However, this result is not clinically significant as there are no reports of evidence documented on the influence of age on pain perception [9,28].

The pre-operative symptom in this study contains some symptoms such as percussion sensitivity, bite pain, periodontal condition, and swelling. The positive and negative of symptom and the diagnosis had no effects on the occurrence of PD and PI. In the previous study, it was reported that patients undergoing single visit root canal treatment with preoperative pain tend to expect and report higher rates of postoperative pain [31]. Therefore, this study shows that root canal obturation after improvement of the patient’s symptoms is not significantly related to the incidence of PD and PI.

The difference of initial- or re-treatment of the tooth had no effects on the occurrence of PD and PI. However, type of endodontic treatment showed a statistically significant difference in the occurrence of PD. A residual analysis identified those specific cells contributing the greatest to the chi-square test result. The analysis showed that infected root canal treated teeth slightly induced “Discomfort” during root canal obturation. Additionally, it was revealed that these “Discomfort” teeth had an opened root apex (data not shown). Tooth with an opened root apex may easily receive a mechanical stimulus by a root canal sealer. In contrast, endodontic treatment did not influence the incidence of PI. Evidence in the previous study of endodontic treatment on PI is inconclusive. The incidence of PI may be attributed to the different treatment methods and obturation materials and techniques.

Root canal obturation techniques were not significantly related to the incidence of PD and PI. The reaction of PD and PI was related to the physiochemical characteristics of CS-BG. Our study showed that the “Pain” level (0.5%) of PI reacted by CS-BG was lower when compared with the other sealers, including eugenol-based, non-eugenol-base, resin-based, other bioceramics-based sealers [2,8,9,10]. Additionally, it is asserted that the irritant effect induced by sealers is gradually reduced with passage of time [32,33]. Eight cases of “Discomfort” and “Pain” levels of PI had no symptoms within 7 days (data not shown). In the previous study, it was reported that eugenol induces a toxic effect [13] and that resin during the initial curing phase shows toxic effects [34]. The other bioceramic-based sealer shows high cytotoxicity in its freshly mixed state because of unstable physical property [35,36]. It is suggested that the low percentage of PD and PI recorded in this study results from the physicochemical stability and biocompatibility of CS-BG. In contrast, it is reported that these sealers do not affect the onset of postoperative pain [8,9,10]. The results of this study are consistent with those of the previous studies [9]. In the future plan, it would be beneficial to compare the incidence of postoperative pain after root canal obturation with CS-BG and other sealers.

## 5. Conclusions

“Discomfort” and “Pain” in the teeth obturated with CS-BG was generally low affecting 1% and 0.5% of the study participants, respectively. The low percentage of PD and PI recorded in this study could be potentially explained by low irritative of CS-BG for the periapical tissue. These clinical results show that CS-BG has excellent biocompatibility for periapical tissue, and it can suppress the distress of patients during root canal obturation. In the incidence of PD according to some parameters, infected root canal treated teeth with an opened root apex slightly induced “Discomfort” during root canal obturation. It is necessary to be careful when root canal with an opened root apex is obturated in most cases. In the future, a prospective clinical study would provide a higher level of evidence in the evaluation of CS-BG in comparison with other sealers.

## Figures and Tables

**Figure 1 ijerph-17-08857-f001:**
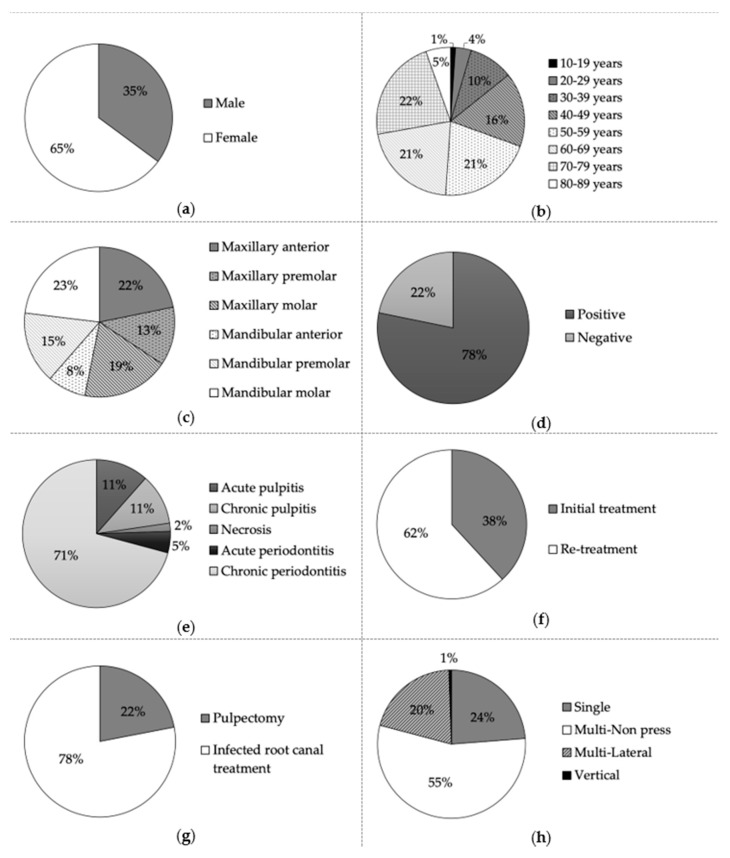
Demographic data: (**a**) Gender; (**b**) Age; (**c**) Arch and type of tooth; (**d**) Symptom; (**e**) Diagnosis; (**f**) Initial- or re-treatment; (**g**) Endodontic treatments; (**h**) Root canal obturation technique.

**Figure 2 ijerph-17-08857-f002:**
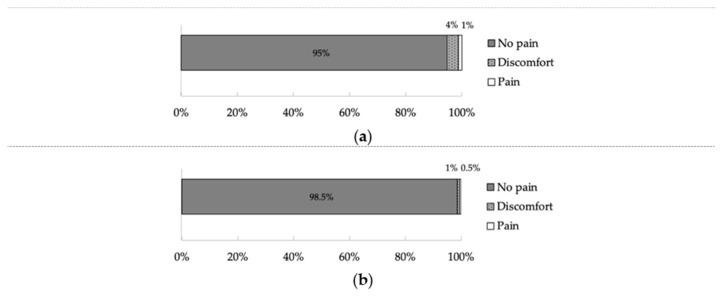
Incidence of pain during root canal obturation (**a**) and pain immediately after root canal obturation (**b**).

**Table 1 ijerph-17-08857-t001:** Incidence of “No pain,” “Discomfort,” and “Pain” of teeth during root canal obturation according to gender, age, arch, type of tooth, symptom, diagnosis, initial- or re-treatment, endodontic treatment, and root canal obturation technique.

Total, *n* = 555		No Pain (*n* = 525)	Discomfort (*n* = 22)	Pain (*n* = 8)	*p* Value
Gender	Male	189	5	1	0.178
	Female	336	17	7	
Age	<50 years	164	2	3	0.079
	≥50 years	361	20	5	
Arch	Maxillary	278	14	3	0.413
	Mandibular	247	8	5	
Type of tooth	Anterior	155	11	1	0.238
	Premolar	148	5	3	
	Molar	222	6	4	
Symptom	Positive	409	17	8	0.893
	Negative	116	5	0	
Diagnosis	Acute pulpitis	59	2	3	0.196
	Chronic pulpitis	59	2	1	
	Necrosis	9	1	0	
	Acute periodontitis	24	1	1	
	Chronic periodontitis	385	16	3	
Initial-/Re- treatment	Initial treatment	198	8	6	0.097
	Re-treatment	327	14	2	
Endodontic treatments	Pulpectomy	205	3	4	0.044 *
	Infected root canal treatment	320	19	4	
Root canal obturation techniques	Single	125	4	3	0.256
	Multi-Non press	285	17	5	
	Multi-Lateral	112	1	0	
	Vertical	3	0	0	

* χ^2^-test (*p* < 0.05).

**Table 2 ijerph-17-08857-t002:** Incidence of “No pain,” “Discomfort,” and “Pain” of teeth immediately after root canal obturation according to gender, age, arch, type of tooth, symptom, diagnosis, initial- or re-treatment, endodontic treatment, and root canal obturation technique.

Total, *n* = 555		No Pain (*n* = 547)	Discomfort (*n* = 5)	Pain (*n* = 3)	*p* Value
Gender	Male	193	3	0	0.431
	Female	354	2	3	
Age	<50 years	168	1	0	0.452
	≥50 years	379	4	3	
Arch	Maxillary	289	4	2	0.430
	Mandibular	258	1	1	
Type of tooth	Anterior	164	2	1	0.507
	Premolar	156	0	0	
	Molar	227	3	2	
Symptom	Positive	426	5	3	0.894
	Negative	121	0	0	
Diagnosis	Acute pulpitis	64	0	0	0.860
	Chronic pulpitis	61	1	0	
	Necrosis	10	0	0	
	Acute periodontitis	26	0	0	
	Chronic periodontitis	386	4	3	
Initial-/Re-treatment	Initial treatment	210	2	0	0.393
	Re-treatment	337	3	3	
Endodontic treatments	Pulpectomy	121	1	0	0.650
	Infected root canal treatment	426	4	3	
Root canal obturation techniques	Single	131	0	1	0.556
	Multi-Non press	300	5	2	
	Multi-Lateral	113	0	0	
	Vertical	3	0	0	

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
