# Peer review of "Effect of Bioactive Glass-Based Root Canal Sealer on the Incidence of Postoperative Pain after Root Canal Obturation"

_ijerph, 2020, doi:10.3390/ijerph17238857_

Round 1

Reviewer 1 Report

I read with great interest the manuscript entitled: "Effect of Bioactive Glass.Based Root canal sealer on the incidence of postoperative pain after root canal obturation" and the topic is very interesting. The study is appropriate and may be suitable for being published in IJERPH.

However, some minor corrections should be performed, for example:

KEYWORDS:

  • Should appear in alphabetical order

INTRODUCTION:

From lines 53-57: The composition of the sealer should be described in Material & Methods, not in Introduction. It would be interesting to add a table with components and %.

Please, add some bibliography in introduction about the range of postoperative pain with the use of different sealears.

MATERIAL AND METHODS:

Please, describe the experience of each of the eleven operators and if you calculated intra-operator and inter-operator statistical differences.

It would be interesting to make a lineal regression to compare the incidence of the different parameters evaluated (age, sex, teeth location, infected/not infected, obturation techniques, etc) on the results.

DISCUSSION:

  • In my view, it is too short. It should be a little more elaborate. Discuss the use of other sealers,.

Author Response

  We greatly appreciate the Reviewer’s valuable comments to improve the manuscript. We have improved the manuscript according to the comments.

Reviewer #1:

The comments

I read with great interest the manuscript entitled: "Effect of Bioactive Glass.Based Root canal sealer on the incidence of postoperative pain after root canal obturation" and the topic is very interesting. The study is appropriate and may be suitable for being published in IJERPH.

However, some minor corrections should be performed, for example:

  1. KEYWORDS: Should appear in alphabetical order

  Based on the Reviewer’s comment, keywords were appeared in alphabetical order at page 1, lines 31, 32.

  1. INTRODUCTION:

2-1. From lines 53-57: The composition of the sealer should be described in Material & Methods, not in Introduction. It would be interesting to add a table with components and %.

  Based on the Reviewer’s comment, the composition of the sealer was described at page 3, lines 98-100 in the Materials and Methods section. Unfortunately, we cannot disclose the rate of components due to the circumstance of manufacturing company.    

2-2. Please, add some bibliography in introduction about the range of postoperative pain with the use of different sealers.

Based on the Reviewer’s comment, one sentences and 3 bibliographies were added at page 1, lines 39, 40 in the Introduction section and the Reference section.

  1. MATERIAL AND METHODS:

3-1. Please, describe the experience of each of the eleven operators and if you calculated intra-operator and inter-operator statistical differences.

Based on the Reviewer’s comment, the experience of each of the eleven operators was added at page 2, lines 71, 72 in the Materials and Methods section. Unfortunately, we did not investigate the association between PP and operators. In the future plan, we would like to investigate the point.

3-2. It would be interesting to make a lineal regression to compare the incidence of the different parameters evaluated (age, sex, teeth location, infected/not infected, obturation techniques, etc) on the results.

  Thank you for your nice advice. We plan to increase the number of samples and continue this study. Therefore, we would like to make a lineal regression in the future work.

  1. DISCUSSION: In my view, it is too short. It should be a little more elaborate. Discuss the use of other sealers.

  Based on the Reviewer’s comment, some sentences and bibliographies were added and modified at pages 7, lines 186-191, 203-215 in the Discussion section and Reference section.

Reviewer 2 Report

The manuscript is well written 

easy to read 

need to improve: 

a) discussion

b) conclusion

c) references

Additional comment:

Each part of the mansucript is well written, but need to:
  • improve references
  • write a precise conclusion
  • re-write the discussion section and compare the study with similars

Author Response

  We greatly appreciate the Reviewer’s valuable comments to improve the manuscript. We have improved the manuscript according to the comments.

Reviewer #2:

The comments

  1. The manuscript is well written

easy to read

need to improve:

  1. a) discussion
  2. b) conclusion
  3. c) references

Additional comment:

Each part of the mansucript is well written, but need to:

  • improve references
  • write a precise conclusion
  • re-write the discussion section and compare the study with similars

  Based on the Reviewer’s comment, some sentences and bibliographies were added at page 7, lines 186-191, 203-215, and Page 8, lines 221-225 in the Discussion section, Conclusion section, and Reference section.

Reviewer 3 Report

This is a very concise but well written paper presented outcome of bioactive glass-based root canal sealer, Nishika Canal Sealer BG (CS-BG), on the incidence of postoperative pain (PP) after root canal obturation (RCO) on large number of teeth, 555.  Results are presented clearly and discussion is also relevant. However, there are some unexplained concerns regarding the methodology.

1- 11 dentists performed root canal therapy. Were all these dentist had the same years of training and expertise, was there any dentist trained as endodontists?

2- The detail diagnosis of teeth prior to root canal therapy (RCT) is not clear, i.e acutes pulpitis, chronic pulpitis , necrosis, acute/chronic periodical lesion, cysts, extraordinaries/intra oral swelling, iatrogenic exposure , irreversible pulpitis, tooth crack syndrome, vital vs non vital.

3- Details pre treatment symptoms are also not clear. Symptoms such as percussion sensitivity , bite pain, periodontal condition, and swelling are important to document and correlate with post operative pain. 

3- Were any of these patients medicated with antibiotics, analgesic, steroids pre or post RCT?

4- How does the result obtained from CS-BG compared to other root canal sealers.  Since there was no control or another standard sealer to make statistical comparisons , how does the result reported here compares to the results available from other root canal sealers?

5- In there a future plan to continue this study with controls or other root canal sealer groups

I suggest she of these concerns to be addressed in results and discussion.

Author Response

  We greatly appreciate the Reviewer’s valuable comments to improve the manuscript. We have improved the manuscript according to the comments.

Reviewer #3:

The comments

This is a very concise but well written paper presented outcome of bioactive glass-based root canal sealer, Nishika Canal Sealer BG (CS-BG), on the incidence of postoperative pain (PP) after root canal obturation (RCO) on large number of teeth, 555.  Results are presented clearly and discussion is also relevant. However, there are some unexplained concerns regarding the methodology.

  1. Eleven dentists performed root canal therapy. Were all these dentists having the same years of training and expertise, was there any dentist trained as endodontists?

Based on the Reviewer’s comment, the experience of each of the eleven dentists was added at page 2, lines 71, 72 in the Materials and Methods section.

  1. The detail diagnosis of teeth prior to root canal therapy (RCT) is not clear, i.e acutes pulpitis, chronic pulpitis, necrosis, acute/chronic periodical lesion, cysts, extraordinaries/intra oral swelling, iatrogenic exposure, irreversible pulpitis, tooth crack syndrome, vital vs non vital.

Based on the Reviewer’s comment, the diagnosis of tooth was added in patient attributes. It was described in the Materials and Methods section, Results section, Discussion section, Figure 1, Table 1, and Table 2.

  1. Details pr-treatment symptoms are also not clear. Symptoms such as percussion sensitivity, bite pain, periodontal condition, and swelling are important to document and correlate with postoperative pain.

In this study, we investigated not detail preoperative pain but the positive and negative of symptom (percussion sensitivity, bite pain, periodontal condition, and swelling et al.). Therefore, it was described in the Materials and Methods section, Results section, Discussion section, Reference section, Figure 1, Table 1, and Table 2.

  1. How does the result obtained from CS-BG compared to other root canal sealers. Since there was no control or another standard sealer to make statistical comparisons, how does the result reported here compares to the results available from other root canal sealers?

  We cannot make a comparison between the result obtained from CS-BG and the result obtained from another sealer. Therefore, we think that it would be beneficial to compare the results of this study with other studies analyzing pain after root canal obturation with different sealers by references.

  Some sentences and some bibliography were added at page 7, lines 203-215 in the Discussion section and the Reference section.

  1. In there a future plan to continue this study with controls or other root canal sealer groups

I suggest she of these concerns to be addressed in results and discussion.

One sentence was added at page 7, lines 214, 215 in the Discussion section.

Reviewer 4 Report

This manuscript presents the effect of bioactive glass-based root canal sealer on postoperative pain. This topic is relevant and presents valuable information for dentists practicing root canal treatment on the recently introduced canal sealer's clinical significance. I have no coments.

the additional comments are listed below:

Was calcium hydroxide used in every case?

It is said it was removed after the improvement of the patient’s symptoms. How long was it?

Was there any minimal time for calcium hydroxide to be left in the root canals?

Was there any correlation between the length of calcium hydroxide present in the canal with the postoperative pain?

It would be beneficial to compare the results of the present study with other studies analysing pain after root canal obturation with different sealers. 

Author Response

  We greatly appreciate the Reviewer’s valuable comments to improve the manuscript. We have improved the manuscript according to the comments.

Reviewer #4:

The comments

This manuscript presents the effect of bioactive glass-based root canal sealer on postoperative pain. This topic is relevant and presents valuable information for dentists practicing root canal treatment on the recently introduced canal sealer's clinical significance.

I have no comments.

the additional comments are listed below:

Was calcium hydroxide used in every case?

Calcium hydroxide was used in every case as a standard treatment.

It is said it was removed after the improvement of the patient’s symptoms. How long was it?

It was from 1week to a few months depending on a case.

Was there any minimal time for calcium hydroxide to be left in the root canals?

About 1 week.

Was there any correlation between the length of calcium hydroxide present in the canal with the postoperative pain?

In this study, we did not investigate the length of calcium hydroxide present in root canal. In the future plan, we would like to investigate the point.

It would be beneficial to compare the results of the present study with other studies analyzing pain after root canal obturation with different sealers.

  Based on the Reviewer’s comment, some sentences and some bibliography were added at page 7, lines 203-215 in the Discussion section and the Reference section.